# Large-scale whole exome sequencing studies identify two genes,*CTSL* and *APOE*, associated with lung cancer

Jingxiong Xu[1], Wei Xu[2,3], Jiyeon Choi[4], Yonathan Brhane[1], David C. Christiani[5], Jui Kothari[6], James McKay[7], John K. Field[8], Michael P. A. Davies[8], Geoffrey Liu[2,3], Christopher I. Amos[9,10], Rayjean J. Hung[1,3], Laurent Briollais[1,3]*

1 Prosserman Centre for Population Health Research, Lunenfeld-Tanenbaum Research Institute, Sinai Health, Toronto, Ontario, Canada, 2 Princess Margaret Cancer Center, University Health Network, Toronto, Ontario, Canada, 3 Dalla Lana School of Public Health, University of Toronto, Toronto, Ontario, Canada, 4 Division of Cancer Epidemiology and Genetics, National Cancer Institute, National Institutes of Health, Bethesda, Maryland, United States of America, 5 T. H. Chan School of Public Health, Harvard University, Boston, Massachusetts, United States of America, 6 Department of Environmental Health, T. H. Chan School of Public Health, Harvard University, Boston, Massachusetts, United States of America, 7 International Agency for Research on Cancer, Lyon, France, 8 Department of Molecular and Clinical Cancer Medicine, The University of Liverpool, Liverpool, United Kingdom, 9 Dan L. Duncan Comprehensive Cancer Center, Department of Medicine, Baylor College of Medicine, Houston, Texas, United States of America, 10 Institute for Clinical and Translational Research, Baylor College of Medicine, Houston, Texas, United States of America

* laurent@lunenfeld.ca

**Data Availability Statement:** The ILCCO data is available at the dbGaP (https://identifiers.org/dbgap:phs000878.v2.p1). UK Biobank data is

## Abstract

Common genetic variants associated with lung cancer have been well studied in the past decade. However, only 12.3% heritability has been explained by these variants. In this study, we investigate the contribution of rare variants (RVs) (minor allele frequency <0.01) to lung cancer through two large whole exome sequencing case-control studies. We first performed gene-based association tests using a novel Bayes Factor statistic in the International Lung Cancer Consortium, the discovery study (European, 1042 cases vs. 881 controls). The top genes identified are further assessed in the UK Biobank (European, 630 cases vs. 172 864 controls), the replication study. After controlling for the false discovery rate, we found two genes, *CTSL* and *APOE*, significantly associated with lung cancer in both studies. Single variant tests in UK Biobank identified 4 RVs (3 missense variants) in *CTSL* and 2 RVs (1 missense variant) in *APOE* stongly associated with lung cancer (OR between 2.0 and 139.0). The role of these genetic variants in the regulation of *CTSL* or *APOE* expression remains unclear. If such a role is established, this could have important therapeutic implications for lung cancer patients.

## Author summary

Lung cancer (LC) is the leading cause of cancer death accounting for 18% of all cancer deaths. Previous studies have suggested genetic contribution to the disease. Common genetic variants associated with LC have been well studied through large, collaborative,

accessible through researcher's application at
www.ukbiobank.ac.uk.

**Funding:** This work was supported by a CIHR
operating grant and NSERC grant (LB, JX). We also
would like to acknowledge funding supports for
ILCCO study: NIH/NCI U01CA209414 (HSPH-
MGH, DCC, JK), NIH U19 CA203654 (Toronto, RH,
GL) and Roy Castle Lung Cancer Foundation
(Liverpool, JKF, MPAD). The funders had no role in
study design, data collection and analysis, decision
to publish, or preparation of the manuscript.

**Competing interests:** The authors have declared
that no competing interests exist.

genome-wide association studies (GWASs) in the past decade. However, they explained
only about 12.3% of LC heritability. It is therefore hypothesized that the unexplained variability might be partially due to rare variants (RVs). In this study, we applied a novel
gene-based test statistic based on a Bayes Factor approach, to whole exome sequencing
data from the International Lung Cancer consortium (ILCCO). Independent replication
of the top genes identified was performed using the UK Biobank data. We found two
genes, *CTSL* and *APOE*, significantly associated with LC in both studies. Within these two
genes, several RVs showed strong associations with lung cancer in the UK Biobank data.
These findings could suggest potential molecular mechanisms leading to lung cancer and
more importantly, possible therapeutic targets for personalized treatment.

## Introduction

Lung cancer (LC) is the most commonly diagnosed cancer in men and the third most commonly occurring cancer in women worldwide as estimated in 2018 [1], with an estimated 2.3
millions new cancers diagnosed annually. It is the leading cause of cancer death worldwide
with 1.8 million annual deaths accounting for 18% of all cancer deaths [1]. Although reduction
of tobacco consumption remains the most appropriate strategy to reduce LC burden, only
10%–15% of all smokers eventually develop LC [2–4]. In Asian countries, up to 30%–40% of
lung cancer cases occur in never smokers [4], which suggests a possible role of genetic factors
among others.

Common genetic variants associated with LC have been identified through large, collaborative, genome-wide association studies (GWASs), including susceptibility loci at *CHRNA3/5*,
*TERT*, *HLA*, *BRCA2*, *CHEK2* [5,6]. Yet, they explained only about 12.3% of LC heritability
reported in a recent GWAS[7]. It is therefore hypothesized that some of the unexplained variability might be due to rare variants (RVs) [8]. A recent study was able to identify 48 germline
RVs with deleterious effects on LC in known candidate genes such as *BRCA*2 in a sample of
260 case patients with the disease and 318 controls [9]. More recently, Liu et al. [10] identified
25 deleterious RVs associated with LC susceptibility, including 13 reported in ClinVar. Of the
five validated candidates, the authors identified two pathogenic variants in known LC susceptibility loci, *ATM* p.V2716A (Odds Ratio 19.55, 95%CI [5.04,75.6]) and *MPZL2* p.I24M frameshift deletion (Odds Ratio 3.88, 95%CI [1.71,8.8]); and three in novel LC susceptibility genes
including *POMC*, *STAU2* and *MLNR*.

To improve the detection of RVs in sequencing studies, we recently proposed a gene-based
test for case-control study designs using a Bayes Factors (BF) statistic [11], comparing the total
RV counts between cases and controls. Informative priors can be included in this setting, making the BF also sensitive to allelic distribution differences at single variant sites between cases
and controls. To elucidate the inherited germline RVs associated with LC, we applied our
novel BF approach to whole exome sequencing (WES) data from the International Lung Cancer consortium (ILCCO) [10], with the goal to identify new genes associated with LC specifically focused on RVs as well as potential causal variants within these genes. Independent
replication of the most promising genes and RVs was performed in the UK Biobank data [12].

## Methods

### Ethics statement

All participants provided written informed consent, and the study was reviewed and approved
by institutional ethic committee of each study site including HSPH-MGH, University Health

Network and Mount Sinai Hospital in Toronto (Toronto), University of Liverpool in UK (Liverpool) and IARC.

## Study population for gene-based and RV discovery

Case patients with LC and matched healthy individuals were identified from four independent case series that form the ILCCO consortium, including Harvard University School of Public Health/Massachusetts General Hospital (HSPH-MGH), University Health Network and Mount Sinai Hospital in Toronto (Toronto), University of Liverpool in UK (Liverpool) and the International Agency for Research on Cancer (IARC). The original data includes 2047 samples, of which 44 are HapMap controls and 68 were flagged by the Center for Inherited Disease Research (CIDR) as duplicates, related individuals or quality control outliers. Whole exome sequencing was performed for selected LC cases and frequency-matched unaffected controls, to identify novel common and rare genetic variants associated with LC risk. To enrich the relevance of genetics in the cases, LC patients were preferentially selected from those with a family history of LC among first-degree relative or early-onset (<60 years). About the same number of controls were selected, frequency-matched by age and sex with the cases. To adjust for population stratification, principal components (PCs) were derived from the genome-wide data from the ILCCO. The analysis was restricted to those with European ancestry. The representation of the top 3 PCs (S1 Fig) identified one outlier participant with possible non-European ancestry, and was removed from the analysis. We further removed 10 individuals with genotype missing rate >10% and one individual was flagged with very low heterozygosity rate (> 6 standard deviations below the mean heterozygosity). After the filtering steps, a total of 1923 subjects remained in the study and were included in the analyses.

## Study population for gene-based and RV replication

We used UK Biobank WES data as the validation set [13,14]. Among the total number of 200,643 samples, our analysis includes all LC patients after excluding those diagnosed at most 5 years before any other primary cancers and controls with no cancer diagnosis history. We also removed at random one individual from each pair of individuals closer than 3rd degree relatives (kinship coefficient > 0.0884), and subjects who self-reported a non-white ethnic background. After the filtering, 173,494 individuals remained in the study.

## Germline Sequencing/QC

**ILCCO.** The sequencing of whole exomes and additional targeted regions of DNA samples from all 4 different sites was performed at the CIDR. Targeted regions were selected based on previous associations with LC or with histological LC subtypes from GWASs on common variants [5,6]. After initial quality control (QC) analysis by CIDR [10], the mean on-target coverage was 52X and more than 97% of targeted bases had a depth greater than 10X. Further QC analysis was performed including the following steps: i) Exclusion of variants with QUAL<100 indicating a low probability that there is a variant at a site or mean GQ<50 indicating low probabilities that genotype calls were correct across individuals at a site so that Ts/Tv ratio is greater than 2 (S2 Fig); ii) Exclusion of singleton variants (variant with occurrence of only 1 minor allele) when minor allele has GQ<50 or depth <20; iii) Exclusion of non-biallelic variants and variants on the sex chromosome; iv) Exclusion of variants with p-value of Hardy-Weinberg equilibrium test <1e-7 in the control samples; v) Set individual genotype as missing if GQ<30 or depth<10; vi) Exclusion of variants with minor allele frequency (MAF)> 1% (MAF was estimated using study population). The MAF distribution of the remaining RVs is given in Table 1.

**Table 1. MAF distribution of genetic variants in the discovery study (ILCCO).**

| MAF | 0 | (0,0.01) | [0.01,0.05) | [0.05,0.5) | Total |
|---|---|---|---|---|---|
| #(Rare Variants) | 136485 | 1022101 | 60288 | 129789 | 1348663 |
| Proportion (%) | 10.12 | 75.79 | 4.47 | 9.62 | 100 |

**UK Biobank.** We performed the following QC steps for all genes selected in the discovery set: i) exclude variants that are not bi-allelic and those with QUAL<10; ii) filter out variants with mean GQ<30 as well as singleton variants with depth <20 or GQ<40; iii) set genotype missing if depth<10 or GQ<20, and exclude variants with missing genotype rate >10%; iv) exclude variants with MAF>1% (MAF estimated using study population).

In both the discovery and replication studies, for our gene-based analyses, we considered -/+ 1k bp up- and down-stream sites of each gene (including non-exonic RVs) for the analysis.

**Gene-based analysis.** To increase the power of discovering genes associated with LC, we applied a gene-based approach based on a Bayes Factor (BF) statistic that we recently developed, to both the discovery and replication studies [11]. It was designed specifically to test the association between a set of RVs located in the same region or in a gene and a disease outcome in the context of case-control designs. An advantage of our BF approach over existing methods is the possibility to introduce an "informative" prior to gain power to detect gene-based associations, where this prior is sensitive to allelic differences between cases and controls for a particular gene (S1 Text). Compared to the commonly-used SKAT gene-based test [15], our BF approach is more sensitive to an excess of small p-values from single RV tests within each gene while SKAT has better power to detect genes exhibiting systematic allelic differences between cases and controls across all RVs. This difference was discussed in details in [11] and illustrated on two genes that showed large discrepancy in overall ranking when applying these two approaches [11]. In this study, we applied two versions of the BF test statistic, $BF_{KS}$ and $BF_{SKAT}$, where either a Kolmogorov-Smirnov (KS) or SKAT p-value is used as informative prior. This gave us higher chance to detect genes that may have different underlying RV allelic distribution differences between cases and controls. The respective advantage of each approach is described in details in the S1 Text. In this paper, we mainly focused on $BF_{KS}$ and used $BF_{SKAT}$ as a secondary analysis.

To assess the sensitivity of the association tests on confounding variables, we conducted sensitivity analyses on the genome-wide significant genes and adjusted our analyses for age, sex, smoking and the top 5 PCs used to control for population stratification. Both the BF and the prior components (KS or SKAT p-value) were adjusted. The extention of $BF_{KS}$ and $BF_{SKAT}$ incorporating covariates is described in S1 Text.

**Single RV-based analysis.** For the two genes that passed a gene-based replication genome-wide significance level (see below), i.e., *APOE* and *CTSL*, we performed single RV tests only with UK Biobank since this study has larger coverage of RVs. We used the Firth's bias-reduced logistic regression to deal with sparse allelic counts [16]. Analyses were adjusted for age, sex, smoking status (ever vs. never smoking) and the top five PCs. RVs that pass a FDR adjusted q value [17] of 0.01 were selected.

**Significance threshold for gene-based replication analysis.** We denote $P_d$ the $P$ value for selecting genes in the discovery cohort (ILCCO) and $P_r$ the $P$ value for selecting a gene in the replication cohort (UK Biobank). We set $\gamma$ as the significance threshold for selecting genes in the discovery cohort and which will be followed-up for replication in UK biobank and $\lambda$ the significance level in the replication cohort. To control the gene-based family-wise error rate

(FWER) $\alpha$, we can determine $\gamma$ and $\lambda$ such that,

$$FWER_{(P_d \leq \gamma, P_r \leq \lambda)} = Pr(V \geq 1) \leq \alpha,$$

where V is number of genes declared achieved signficiance levels in both discovery and validation studies, $P_d \leq \gamma$ and $P_r \leq \lambda$, where $\gamma$ and $\lambda$ were determined through permutation analysis, as follows. First, we repeated analyses of ILCCO (discovery set) and UK Biobank (validation set) studies 100 times, where each time the phenotype of individuals was permuted. Second, we determined the two thresholds such that among 100 replicates, the number of identified significant genes is less or equal to $100 \times \alpha = 100 \times 0.05 = 5$, for a genome-wide control of FWER$\leq$5%. We found the following thresholds, $\gamma = 5 \times 10^{-4}$ and $\lambda = 0.05$ in the discovery and validation study, respectively, when using $BF_{KS}$ as the test statistic (i.e., our main statistic). Therefore, in our application analysis, the set of genes that passed a significance threshold of $\gamma = 5 \times 10^{-4}$ in the discovery (ILCCO) cohort and $\lambda = 0.05$ in the replication (UK Biobank) cohort were declared associated with the disease and replicated.

## Results

### Characteristics of patients in the discovery and replication studies

Our discovery study (ILCCO) includes 1042 lung cancer cases and 881 controls (HSPH-MGH, 426 cases and 270 controls; Toronto, 259 cases and 258 controls; Liverpool, 64 cases and 69 controls; IARC, 293 cases and 284 controls). The replication study (UK Biobank) includes a total of 630 cases and 172,864 controls. In the discovery study, the distributions of sex and age are comparable between cases and controls. However, in the replication study, there is an excess of males in cases compared to controls (52.7% vs. 45.2%, $P = 1.9 \times 10^{-4}$) and cases are older age at enrollment compared to controls (mean = 62.0 vs. 56.7 years, $P < 2.2 \times 10^{-16}$) (Table 2). As expected, there is a higher proportion of never smokers in controls compared to cases (35.2% vs. 11.8% $P < 2.2 \times 10^{-16}$ and 54.6% vs. 14.8% $P < 2.2 \times 10^{-16}$ in the discovery and replication study, respectively).

### Gene-Based analysis

In the discovery study, a total of 13,872 genes with at least 20 bi-allelic RVs were analyzed based on the QC pipeline described. The QQ plots corresponding to 2log($BF_{KS}$) and 2log

**Table 2. Basic demographic characteristics in the discovery and validation studies.**

|  | Discovery (ILCCO) | | | Replication (UK Biobank) | | |
|---|---|---|---|---|---|---|
|  | controls | cases |  | controls | cases |  |
|  | n = 881 | n = 1042 | p-value | n = 172864 | n = 630 | p-value |
| Sex, No. (%) |  |  | NS |  |  | 1.9E-04 |
| M | 513 (58.2) | 613 (58.8) |  | 78163 (45.2) | 332 (52.7) |  |
| F | 368 (41.8) | 429 (41.2) |  | 94701 (54.8) | 298 (47.3) |  |
| Age, mean (SD) | 60.8 (11.8) | 62.2 (12.3) | NS | 56.7 (8.0) | 62.0 (5.8) | <2.2E-16 |
| Smoking, No. (%) |  |  | <2.2E-16 |  |  | <2.2E-16 |
| Never | 310 (35.2) | 123 (11.8) |  | 94378 (54.6) | 93 (14.8) |  |
| Former | 375 (42.6) | 421 (40.4) |  | 61770 (35.7) | 319 (50.6) |  |
| Current | 193 (21.9) | 492 (47.2) |  | 16119 (9.3) | 214 (34.0) |  |
| Missing | 3 (0.3) | 6 (0.6) |  | 597 (0.3) | 4 (0.6) |  |

NS: not significant

($BF_{SKAT}$) statistics are presented in Fig 1 and confirm that they are both asymptotically distributed as $\chi^2(3)$. Using a significance level of $\gamma = 5\times10^{-4}$ in the discovery cohort (see Methods section), a total of 17 genes based on $BF_{KS}$ and 14 genes using $BF_{SKAT}$ (Tables 3 and 4) were selected for replication. The 2 top genes are *CTSL* ($P = 4.9\times10^{-5}$) and *TBX4* ($P = 6.5\times10^{-5}$) with $BF_{KS}$, *VAV2* ($P = 1.9\times10^{-5}$) and *DENND4B* ($P = 4.3\times10^{-5}$) with $BF_{SKAT}$. Four genes are found by both test statistics including *CTSL*, *TBX4*, *C8orf44*, and *DGKB*. Using a significance level of $\lambda = 0.05$ (see Methods section) in the replication study, we were able to replicate only one gene, *CTSL* ($P = 2.7\times10^{-3}$), when using the $BF_{KS}$ test and the two genes *APOE* ($P = 1.9\times10^{-3}$) and *CTSL* ($P = 6.9\times10^{-6}$) based on the $BF_{SKAT}$ test (Tables 3 and 4). For each gene identified in the discovery set, we calculated an overall p-value in Tables 3 and 4 by combining p-values from the discovery and validation sets using Fisher's method [18].

## Sensitivity analysis

We found that the association signal for *CTSL* did not change much after adjustment for confounders using $BF_{KS}$ (unadjusted: discovery p-value = 4.87E-05, validation p-value = 2.75E-03; adjusted: discovery p-value = 2.84E-05, validation p-value = 3.88E-03) (S1 Table) and $BF_{SKAT}$ (unadjusted: discovery p-value = 4.30E-04, validation p-value = 1.31E-05; adjusted: discovery p-value = 1.32E-03, validation p-value = 4.33E-05) (S2 Table). While the adjusted association using $BF_{SKAT}$ on *APOE* (discovery p-value = 2.12E-03, validation p-value = 8.24E-03) (S2 Table) was not as significant as the unadjusted $BF_{SKAT}$ (discovery p-value = 2.56E-04, validation p-value = 4.01E-03). Of note, in this analysis, 9 out of 1923 individuals were removed from ILCCO study due to the missing smoking status and 761 out of 173,494 individuals were removed from UK Biobank study due to the missing values of smoking and/or PCs.

## Single RV-based analysis

In UK Biobank, a total of 155 bi-allelic RVs for *CTSL* and 174 for *APOE* were included in the analysis. In *CTSL*, 4 RVs were found associated with LC at an FDR q-value of 0.01, including

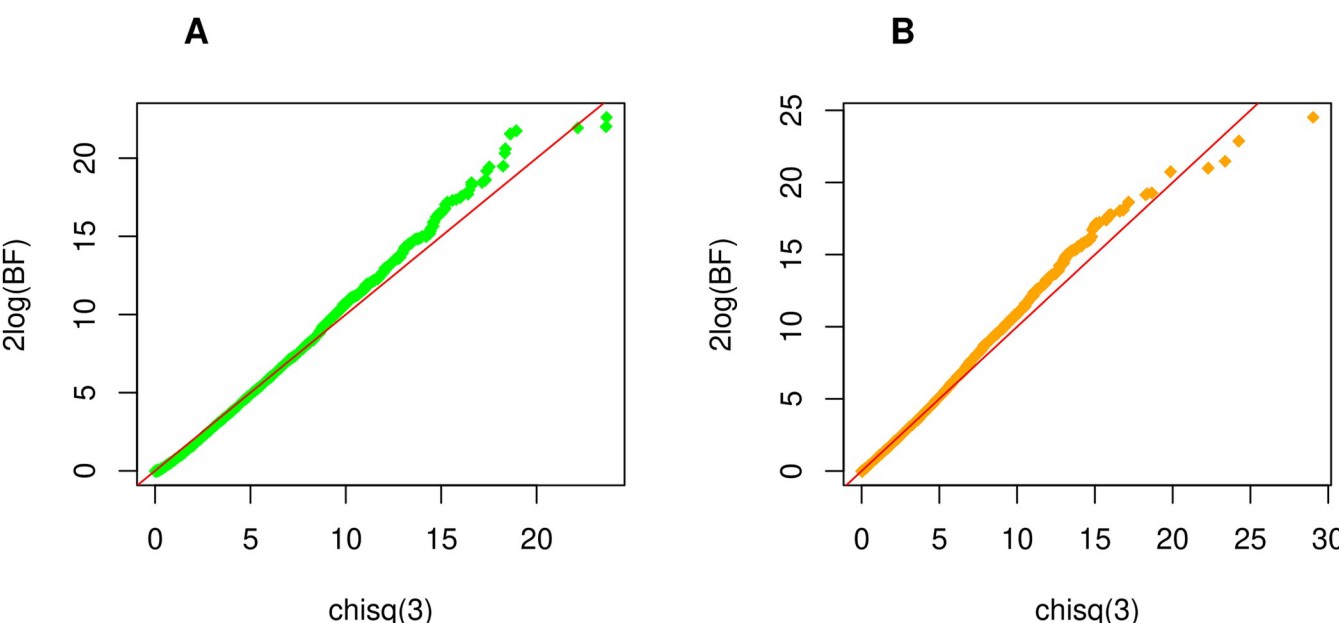

**Fig 1. QQ plot of ILCCO WES study.** The departure of the right tail from the 45 degree line represents the association signals from the study. (A) illustrates results using BF with KS prior. Under the null hypothesis (no association between genes and phenotype), $2\log BF_{ks} \sim \chi^2(3)$. (B) shows results using BF with SKAT prior. Similarly, $2\log BF_{SKAT} \sim \chi^2(3)$ under the null hypothesis.

**Table 3. Results of gene-based analyses using BF$_{KS}$ test[a] in the discovery and replication studies.**

| Rank | Genes | Chr | #(Sites) | Discovery (ILCCO) | | Replication (UK Biobank) | | Combined P |
|---|---|---|---|---|---|---|---|---|
| | | | | KS P[b] | BF$_{KS}$ P[c] | KS P[b] | BF$_{KS}$ P[c] | Fisher's method |
| 1 | CTSL | 9 | 25 | 1.32E-03 | **4.87E-05** | 8.43E-01 | **2.75E-03** | **2.26E-06** |
| 2 | TBX4 | 17 | 37 | 1.48E-03 | 6.49E-05 | 9.67E-01 | 9.96E-01 | 6.88E-04 |
| 3 | RASL10B | 17 | 53 | 4.05E-04 | 6.75E-05 | 1.00E+00 | 9.81E-01 | 7.03E-04 |
| 4 | MUC3A | 7 | 94 | 1.30E-04 | 7.33E-05 | 5.95E-01 | 6.42E-01 | 5.16E-04 |
| 5 | AMN | 14 | 22 | 1.68E-04 | 8.08E-05 | 9.07E-01 | 9.71E-01 | 8.20E-04 |
| 6 | KRTAP19-4 | 21 | 21 | 3.38E-05 | 1.27E-04 | 8.74E-02 | 8.76E-02 | 1.38E-04 |
| 7 | KRTAP19-5[d] | 21 | 20 | 3.38E-05 | 1.28E-04 | NA | NA | NA |
| 8 | CPB2 | 13 | 25 | 1.74E-03 | 1.46E-04 | 1.00E+00 | 6.71E-01 | 1.01E-03 |
| 9 | C8orf44 | 8 | 38 | 1.11E-02 | 2.17E-04 | 6.74E-01 | 1.82E-01 | 4.39E-04 |
| 10 | ZW10 | 11 | 48 | 6.49E-04 | 2.23E-04 | 8.79E-01 | 7.19E-01 | 1.56E-03 |
| 11 | INHA | 2 | 68 | 1.05E-04 | 2.51E-04 | 1.00E+00 | 9.04E-01 | 2.13E-03 |
| 12 | DGKB | 7 | 79 | 3.33E-02 | 3.27E-04 | 9.73E-01 | 9.34E-01 | 2.77E-03 |
| 13 | FBXO6 | 1 | 55 | 2.09E-03 | 3.34E-04 | 3.47E-01 | 3.51E-01 | 1.18E-03 |
| 14 | PHF12 | 17 | 82 | 2.00E-03 | 3.57E-04 | 1.00E+00 | 8.35E-01 | 2.71E-03 |
| 15 | LEMD3 | 12 | 46 | 8.70E-04 | 3.58E-04 | 1.00E+00 | 9.98E-01 | 3.20E-03 |
| 16 | OR5AC2 | 3 | 70 | 1.09E-04 | 3.85E-04 | 1.00E+00 | 1.00E+00 | 3.41E-03 |
| 17 | FGF8 | 10 | 38 | 9.89E-02 | 4.52E-04 | 9.93E-01 | 7.29E-01 | 2.97E-03 |

a. Bayes factor (BF) approach using Kolmogorov-Smirnov (KS) test as prior

b. P value of KS test

c. P value of BF with KS prior

d. Genes with #(sites)<20 were excluded from BF test

**Table 4. Results of gene-based analyses using BF$_{SKAT}$[a] in the discovery and replication studies.**

| Rank | Genes | Chr | #(Sites) | Discovery (ILCCO) | | Replication (UK Biobank) | | Combined P |
|---|---|---|---|---|---|---|---|---|
| | | | | SKAT P[b] | BF$_{SKAT}$ P[c] | SKAT P[b] | BF$_{SKAT}$ P[c] | Fisher's method |
| 1 | VAV2 | 9 | 121 | 3.09E-04 | 1.95E-05 | 6.72E-01 | 5.72E-01 | 1.39E-05 |
| 2 | DENND4B | 1 | 69 | 2.21E-05 | 4.31E-05 | 9.96E-01 | 6.35E-01 | 3.15E-04 |
| 3 | TBX4 | 17 | 37 | 1.95E-03 | 8.41E-05 | 8.21E-01 | 9.41E-01 | 8.27E-04 |
| 4 | RHBDL3 | 17 | 27 | 9.09E-03 | 1.06E-04 | 1.63E-01 | 2.91E-01 | 3.51E-04 |
| 5 | C8orf44 | 8 | 38 | 5.89E-03 | 1.19E-04 | 9.97E-01 | 2.52E-01 | 3.43E-04 |
| 6 | CCT8 | 21 | 46 | 2.43E-02 | 2.41E-04 | 9.87E-01 | 9.99E-01 | 2.25E-03 |
| 7 | SIGLEC11 | 19 | 24 | 3.10E-03 | 2.46E-04 | 7.23E-01 | 5.81E-01 | 1.41E-03 |
| 8 | APOE | 19 | 25 | 2.65E-04 | **2.56E-04** | 6.10E-03 | **4.01E-03** | **1.52E-05** |
| 9 | POMK | 8 | 33 | 3.00E-02 | 3.27E-04 | 9.54E-01 | 7.50E-01 | 2.29E-03 |
| 10 | DGKB | 7 | 79 | 4.34E-02 | 4.20E-04 | 3.79E-01 | 5.10E-01 | 2.02E-03 |
| 11 | CTSL | 9 | 25 | 1.29E-02 | **4.30E-04** | 3.08E-03 | **1.31E-05** | **1.13E-07** |
| 12 | CPB2 | 13 | 25 | 5.55E-03 | 4.42E-04 | 2.98E-01 | 2.65E-01 | 1.18E-03 |
| 13 | ITGB6 | 2 | 61 | 3.23E-02 | 4.93E-04 | 9.83E-01 | 9.40E-01 | 4.02E-03 |
| 14 | VCPIP1 | 8 | 39 | 1.73E-02 | 4.94E-04 | 8.73E-01 | 7.00E-01 | 3.10E-03 |

a. Bayes factor (BF) approach using SKAT as prior

b. P value of SKAT test

c. P value of BF with SKAT prior

**Table 5. Results of single RV-based association analysis in the genes *CTSL* and *APOE* using UK Biobank data.**

| Gene (Variant, position) | ClinVar Significance [19] | Overall (N = 173,494) | | Cases (N = 630) | | Controls (N = 172,864) | | Association | | |
|---|---|---|---|---|---|---|---|---|---|---|
| | | MAF | #Carriers | MAF | #Carriers | MAF | #Carriers | P value[a] | FDR q-value[b] | Odds Ratio[a] (95% CI) |
| *CTSL* (rs771328780, chr9: 87,728,433) | Unknown | 4.0E-5 | 14 | 1.6E-03 | 2 | 3.5E-05 | 12 | 6.7E-5 | 7.1E-4 | 83.9 (18.2–387.2) |
| *CTSL* (rs778002071, chr9: 87,729,621) | Missense | 2.0E-05 | 7 | 7.9E-4 | 1 | 1.7E-05 | 6 | 8.0E-4 | 4.3E-3 | 139.0 (20.7–933.7) |
| *CTSL* (rs777251059, chr9: 87,730,426) | Missense | 1.4E-05 | 5 | 7.9E-04 | 1 | 1.2E-05 | 4 | 3.9E-3 | 9.8E-3 | 54.8 (7.8–382.6) |
| *CTSL* (rs112682750, chr9: 87,727,608) | Missense | 7.8E-03 | 2694 | 1.5E-02 | 19 | 7.7E-03 | 2675 | 7.8E-03 | 0.01 | 2.0 (1.3,3.1) |
| *APOE* (chr19: 44,907,893) | Unknown | 1.2E-05 | 4 | 7.9E-04 | 1 | 8.7E-06 | 3 | 2.8E-4 | 5.5E-3 | 276.3 (38.5–1985.3) |
| *APOE* (rs1568615382 chr19: 44,906,640) | Missense | 3.2E-05 | 11 | 7.9E-04 | 1 | 2.9E-05 | 10 | 1.4E-4 | 0.01 | 90.0 (15.5–523.9) |

[a]Based on the Firth biased-corrected logistic regression [16]
[b]Only RVs with a q-value ≤ 0.01 were selected.

variant at positions 87728433 (rs771328780), 87729621 (rs778002071), 87730426 (rs777251059) and 87727608 (rs112682750) on chromosome 9 (Table 5), where the last 3 were missense variants. In *APOE*, 2 RVs passed this significance level, including variant at position 44907893 (rs number not available) and 44906640 (rs1568615382) on chromosome 19. Most of the variants found to be associated with LC risk are very rare (MAF$<10^{-4}$ in controls), except one missense variant in *CTSL*, rs112682750, has a MAF of $7.7 \times 10^{-3}$.

All the 6 RVs are associated with increased LC risk as indicated by an odds-ratio>1 in UK Biobank. One of the 6 RVs was present in ILCCO, rs112682750 in *CTSL*, but it did not show association with LC after adjustment for age, sex, smoking and PCs ($P = 0.19$).

## Genomic region analysis of rs112682750 in *CTSL*

Using cancer cell lines from the USCS genome browser, a genomic analysis of the region around rs112682750 indicates that this variant is located within a promoter/enhancer region of *CTSL* in lung related cells (S3 Fig). This suggests that rs112682750 might affect the transcription of *CTSL*.

## Annotation of Single RVs in *CTSL* and *APOE*

We searched functional annotation for the 6 associated RVs identified from *CTSL* and *APOE* using Ensembl Variant Effect Predictor (VEP) [20], Combined Annotation Dependent Depletion (CADD) [21,22] and Functional Annotation of Variants–Online Resource (FAVOR) [23]. The search results indicated that rs778002071 (*CTSL*) was categorized as deleterious nonsynonymous variant, according to all three annotation resources, and the rest 5 RVs were predicted to be tolerated (benign) by at least one resource (Table 6).

## Discussion

By focusing on rare variants using whole exome sequencing data, we identified two new genes, *CTSL* and *APOE*, associated with LC in the ILCCO study, that were replicated in the UK

**Table 6. Functional annotation of rare variants in the genes *CTSL* and *APOE*.**

| SNP | Allele | Amino acids | Codons | PolyPhen Category[a] | Val[b] | SIFT Category[c] | Val[d] | Category | aPC-Protein-Function[e] PHRED | Percentile | CADD[f] PHRED |
|---|---|---|---|---|---|---|---|---|---|---|---|
| rs771328780 (*CTSL*, 87,728,433) | G | - | - | - | - | - | - | intronic | 2.97 | - | 3.90 |
| rs778002071 (*CTSL*, 87,729,621) | A | G/S | Ggc/Agc | possibly damaging | 0.861 | deleterious | 0.02 | exonic, nonsynonymous | 28.03 | 0.16 | 26.10 |
| rs777251059 (*CTSL*, 87,730,426) | C | G/A | gGt/gCt | benign | 0.059 | tolerated | 0.33 | - | - | - | 21.60 |
| rs112682750 (*CTSL*, 87,727,608) | C | N/T | aAt/aCt | benign | 0.001 | tolerated | 0.99 | exonic, nonsynonymous | 22.17 | 0.61 | 15.00 |
| - (*APOE*, 44,907,893) | A | Q | caG/caA | - | - | - | - | - | - | - | 3.97 |
| rs1568615382 (*APOE*, 44,906,640) | G | A/T | Gct/Act | Possibly damaging | 0.536 | tolerated | 0.09 | - | - | - | 22.9 |

a. PolyPhen category of change [19].

b. PolyPhen score: It predicts the functional significance of an allele replacement from its individual features. Range: [0, 1] (default: 0) [19].

c. SIFT category of change [24].

d. SIFT score, ranges from 0.0 (deleterious) to 1.0 (tolerated). Range: [0, 1] (default: 1) [24].

e. Protein function annotation PC: the first PC of the standardized scores of "SIFTval, PolyPhenVal, Grantham, Polyphen2_HDIV_score, Polyphen2_HVAR_score, MutationTaster_score, MutationAssessor_score" in PHRED scale. Range: [2.974, 86.238] [23].

f. The CADD score in PHRED scale (integrative score). A higher CADD score indicates more deleterious. Range: [0.001, 84] [21,22].

Biobank study. In *CTSL*, 3 missense RVs and 1 RV with unknown significance were discovered as associated with LC in the UK Biobank study. In *APOE*, 1 missense variant and 1 with unknown significance were discovered.

The Cathepsin L gene (*CTSL*), is a ubiquitously expressed lysosomal endopeptidase that is primarily involved in terminal degradation of intracellular and endocytosed proteins [25]. *CTSL* has recently gained attentions for its roles in SARS-CoV2 entry to host cell by cleaving receptor-bound viral spike protein, which results in further activation and infection[26,27]. While potential functional connection between viral infection and lung cancer susceptibility remains to be established, *CTSL* also has roles relevant in tumorigenesis and progression. *CTSL* upregulation has been reported in a wide range of human malignancies including ovarian, breast, prostate, lung, gastric, pancreatic and colon cancers [28]. Importantly, evidence indicates that *CTSL* expression may be linked to cancer grade and stage. In LC patients, higher *CTSL* activity has been reported compared to non-malignant tissue as well as association between tumor grade and upregulated serum levels [29]. The role of *CTSL* in promoting tumor progression and metastatic aggressiveness has also been suggested [30]. Significant interest in the development of *CTSL* intervention strategies has also emerged. For example, *CTSL* downregulation through RNA interference in different tumor models (including glioma, osteosarcoma, myeloma and melanoma) resulted in consistent inhibition of tumorigenicity and invasiveness of neoplastic cells [31–34]. The identification of patients who might benefit from anti-CTSL therapy remains an important clinical question. The identification of new RVs that correlate with LC risk in our study could therefore help identify these patients. Although the impacts of these variants to CTSL levels or activity in early vs. late stages of lung tumorigenesis need to be established, potential regulatory function of the most common variant we identified in *CTSL*, rs112682750, for instance, could be hypothesized.

The apolipoprotein E gene (*APOE*) codes for a protein associated with lipid particles, that mainly functions in lipoprotein-mediated lipid transport between organs via the plasma and interstitial fluids. *APOE* is also associated with atherosclerogenesis, which itself has been involved in tumor development. *APOE* has been shown to act as a growth factor that can influence carcinogenesis [35]. In patients with LC, the levels of *APOE* gene expression were significantly higher in cancer tissue than in adjacent non-cancer tissue [36]. Serum *APOE* has also been associated with lymph node metastasis in lung adenocarcinoma patients [37]. It was also reported that high expression of *APOE* promotes cancer cell proliferation and migration and contributes to an aggressive clinical course in patients with lung adenocarcinoma [38]. *APOE* has also raised interest for therapeutic interventions. For instance, *APOE* was involved in the inhibition of melanoma metastasis and angiogenesis by stimulating the immune response to tumor cells [39]. Identification of genetic variants that could regulate *APOE* expression could therefore have important therapeutic implications. Of note, *APOE* was only detected with one version of our BF approach (i.e., $BF_{SKAT}$) and further validation of this gene is warranted.

The strengths of our study include the large sample sizes available for discovery and replication of the gene-based analyses and the use of UK Biobank data for RV discoveries. Our statistical approach for gene discovery, the Bayes Factor statistic, has also been shown to have increased power compared to competing approaches such as SKAT and the Burden test [11]. Another significant advantage is its sensitivity to detect single RV associations through the definition of informative priors. Under our statistical framework, the discovery of RVs can therefore be thought as a two-step approach where the first step is a gene-based analysis and the second step, an RV association test within the set of significantly associated genes.

Our study contrasts with Liu et al.'s analysis of the ILLCO data [10] in several aspects. They performed single RV analyses focusing only on suspected deleterious variants. In a second step, they performed gene-based tests using only genes that included RVs that were significantly associated with LC after controlling for multiple comparisons from a Burden test. In comparison, we tested all the genes in the discovery cohort and did not make any assumption regarding the possible functional effect of the RVs.

The discovery of RVs in the context of sequencing studies remains a field of intensive research.

The limitations of this study include the need for further validation and characterization of the two genes and RVs identified, in particular to correlate them with disease progression outcomes and LC subtypes. Also, the benefit for therapeutic interventions may be considered as it could lead to a more personalized treatment of LC patients targeting specific gene/pathway mechanisms such as the immune response system.

## Supporting information

**S1 Text. Method Supplement.**
(DOCX)

**S1 Table. Results of gene-based analysis using adjusted $BF_{KS}$ test in the discovery and replication.**
(DOCX)

**S2 Table. Results of gene-based analysis using adjusted $BF_{SKAT}$ test in the discovery and replication.**
(DOCX)

**S1 Fig. Population Structure shown in top 3 principal components.**
(DOCX)

**S2 Fig. Relationship between QUAL and mean GQ vs. Ts/Tv ratio.**
(DOCX)

**S3 Fig. Genetic region of rs112682750 (pos: 87727608, build 38) within *CTSL* gene.**
(DOCX)

## Author Contributions

**Conceptualization:** Jingxiong Xu, Rayjean J. Hung, Laurent Briollais.

**Data curation:** Yonathan Brhane.

**Formal analysis:** Jingxiong Xu, Jiyeon Choi, Yonathan Brhane.

**Investigation:** Jingxiong Xu, David C. Christiani, James McKay, John K. Field, Geoffrey Liu, Christopher I. Amos, Rayjean J. Hung, Laurent Briollais.

**Methodology:** Jingxiong Xu, Wei Xu, Laurent Briollais.

**Supervision:** Laurent Briollais.

**Writing – original draft:** Jingxiong Xu, Laurent Briollais.

**Writing – review & editing:** Jingxiong Xu, Wei Xu, Jiyeon Choi, David C. Christiani, Jui Kothari, James McKay, John K. Field, Michael P. A. Davies, Geoffrey Liu, Christopher I. Amos, Rayjean J. Hung, Laurent Briollais.

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
