## [Decision Letter · Decision Letter 0]

16 Apr 2023

Dear Dr Briollais,

Thank you very much for submitting your Research Article entitled 'Large-scale whole exome sequencing studies identify two genes,CTSL and APOE, associated with lung cancer' to PLOS Genetics.

The manuscript was fully evaluated at the editorial level and by independent peer reviewers. The reviewers appreciated the attention to an important problem, but raised some substantial concerns about the current manuscript. Based on the reviews, we will not be able to accept this version of the manuscript, but we would be willing to review a much-revised version. We cannot, of course, promise publication at that time.

From my editorial perspective, I read the Appendix section detailing the maths behind the test but I could not manage to get an intuitive understanding of the differences between the BF test and the more traditional options, for example the commonly used SKAT test. Additional background in the introduction or the methods sections to explain in easy to understand terms (a toy example exemplifying a situation where the tests diverge perhaps?) what the differences are (and based on Tables 3 and 4 it is substantial) would be something I would like to see in a revised version. I also like the idea of reviewer 1 of a permutation analysis to understand the level of noise in the analyses (which would in turn justify the choice of thresholds). Given that the results are not extremely significant, it seems to me that it is important to be as transparent as possible about the expected distribution of the test statistic.

If you decide to revise the manuscript for further consideration at PLOS Genetics, please aim to resubmit within the next 60 days, unless it will take extra time to address the concerns of the reviewers, in which case we would appreciate an expected resubmission date by email to plosgenetics@plos.org.

We are sorry that we cannot be more positive about your manuscript at this stage. Please do not hesitate to contact us if you have any concerns or questions.

Yours sincerely,

Vincent Plagnol

Academic Editor

PLOS Genetics

David Kwiatkowski

Section Editor

PLOS Genetics

Reviewer's Responses to Questions

**Comments to the Authors:**

Reviewer #1: Xu et al perform a gene-based analysis of whole exome sequence data for lung cancer. They first analyse data from the International Lung Cancer Consortium (1000 cases, 900 controls), then from UK Biobank (600 cases, 170k controls). They claim to identify two genes significantly associated with lung cancer risk, which combined contain 6 associated rare variants.

This is an interesting study, and it appears that the ILCC WES data are the largest WES data available for lung cancer (I could not find larger based on a quick Google). I note that the original analysis of these data (Ref 10) had limited success (only 5 putatively significant single-SNP associations, and no significant genes), so it would be good if a reanalysis using a more powerful tool produced significant findings.

In general, I feel the gene-based results are valid (i.e., that two genes achieve global significance). I disagree that within these genes, 6 individual SNPs are significant (because I disagree with using the FDR to determine significance). However, this is not very concerning, because the single-SNP analysis is only a secondary analysis.

Note that the authors often talk about controlling the false discovery rate (FDR), which initially made me skeptical (because FDR-based significance thresholds can be too relaxed). However, I believe that in practice, their "global FDR" threshold is equivalent to controlling the family-wide error rate (FWER), which is standard. If I am correct, I suggest referring to the threshold as a FWER to avoid confusion.

Major comments

1 - As I say above, I was initially concerned about the significance thresholds used for the gene-based test. However, it seems to me that setting alpha=.05/13872 in Equation on Page 9 corresponds to ensuring the FWER from the meta analysis (i.e., combining gene-based p-values from discovery and replication data) is not higher than 0.05. In fact, your thresholds may be slightly too strict, because your replication analysis considers only a subset of genes (although I do not suggest relaxing to compensate!). Similarly, at first read, I thought the choices of per-study thresholds (i.e., 6e-4 and 5e-2) were suspicious. For example, why pick 6e-4 as the discovery threshold, and not 6e-5? However, I believe changing the threshold will not lead to false positives (because utlimately they do not change the meta analysis threshold), and can instead only lead to false negatives (because overly strict choice of discovery threshold will mean you take no genes forward to replication). Lastly, another similar point is that because you seek to replicate genes that exceed the significance threshold for EITHER the KS OR SKAT-O test, there is a multiple testing issue. However, again, I believe the high correlation between the two tests means the issue is slight, and this is evidenced by the similarity of the results from each.

So in summary, I think more appropriate to say you are fixing the FWER, rather than the global FDR. Further, it would be good to give some justification for 6e-4 discovery threshold (even if if you chose it only for convenience or logistical reasons). Lastly, I would focus on one test, then use the other for a sensitivity analysis (else consider correcting for the fact you are performing multiple tests).

2 - I would be reassured to see results from a permutation analysis. For example, you could repeat the analyses of ILCC and UKBB 100 times, each time permuting the order of individuals (perhaps using residuals adjusted after allowing for covariates), then record how often one or more genes achieve genome-wide significance - ideally, this should happen at most 5 times (if your control of FWER is effective)

Minor comments

3 - I thought the study performed careful QC and it was well explained.

4 - I do not fully understand the equation on page 9. Nonetheless, the resulting thresholds (6e-4 and 5e-3) seem reasonable, because their product is close to 3.6e-6.

5 - I suggest including combined p-values in Tables 3 and 4, eg use Fishers method to combine p-values from discovery and replication analyses.

6 - I do not like that your significance threshold for the single RV analysis is FDR q <0.01 (again, because I am not in favour of the using FDR significance thresholds). I would prefer instead using P<0.05/(155+174)=1.5e-5. While it appears no single SNPs achieve this threshold, I do not consider this a major problem, as this is only a secondary analysis intended to give a better understanding of the main results.

7 - I like that you provide a summary of the key maths of the gene-based method in the supplement.

8 - In the abstract / intro, you discuss how common snp gwas have only found 12.3% of LC heritability, giving the impression that rare variants might explain a large share of LC heritability. However, you do not seem to return to this point. Do you think your results suggest they do not (because your two genes have such small effects)? Or maybe the limited sample size (while big for sequence data, an effective sample size of about 4,000 is small for GWAS) means the contribution of rare variants for LC remains an open question?

Signed Doug Speed

Reviewer #2: This paper reports gene-based association tests using a novel Bayes Factor statistics in ILCCO (a discovery study) and UK Biobank (a replication study). However, a major concern with this method is its inability to control for covariates. Table 2 shows that smoking behavior differs significantly between cases and controls, which is to be expected. However, the Bayes Factor method cannot adjust for this confounding factor. Additionally, other potential confounders, such as top PCs, age, and gender, can compromise the reliability of the analysis results, even though the authors attempted to make these factors comparable between cases and controls. Therefore, it would be advisable for the authors to use established packages like SAIGE-GENE / SAIGE-GENE+ for analysis until the Bayes Factor method can accommodate covariates, as discussed in the Biometrics paper.

As a minor suggestion, adding p-values in Table 2 instead of only in the text could enhance clarity.

Reviewer #3: This paper utilized Bayes Factor statistic to identify RVs associated with lung cancer using ILCOO as the discovery dataset and UK biobank as the replication dataset. The analysis approach is justifiable and the authors found CTSL and APOE were associated with lung cancer in both studies.

My comments on the paper are as follows:

1. Please explain how you identified European ancestry individuals in ILCOO data?

2. I wonder whether the authors considered the batch effects across different sites in ILCOO data.

3. Please specify the specific number of individuals of ILCCO and UK Biobank WES data before and after removing individuals.

4. I wonder how the authors identified rare variants. What if the MAF of a specific variant is >0.01 in ILCCO and <0.01 in UK Biobank.

5. What if a specific RV only be found in one dataset? Did you remove it or keep it?

6. BF statistics are quite interesting. However, SAIGE-GENE and REGENIE are more popular in gene-based tests. Have you compared your results with the other two methods? Btw, both methods I mentioned can control family relatedness.

7. Could you please explain why the choice of lambda (0.005) is larger than gamma (6E-4)? The sample size of UK Biobank is much larger than ILCOO. It will be very helpful if the authors can explain why such a choice of lambda and gamma is optimal.

8. The qq plots (Fig 1) look great. Could you please also show qq plots from validation cohorts? When case control ratio is unbalanced, how does the qq plot of BF statistics look like?

9. Could you explain a little bit about KS and SKAT prior? Which one do the author recommend?

10. I am curious how the authors derived p-values from Bayes factor? P-values are usually not available in Bayesian methods.

11. Firth biased-corrected logistic regression may not perform well when MAC is too small. It’s hard to believe these p-values (except rs112682750) in Table 5 given # carriers. I will suggest the authors perform resampling methods or efficient resampling when # carriers is less than 20.

12. I saw you mentioned data source in Data Availability section. Please add it in the manuscript as well.

**Have all data underlying the figures and results presented in the manuscript been provided?**

Reviewer #1: Yes

Reviewer #2: None

Reviewer #3: Yes

PLOS authors have the option to publish the peer review history of their article (what does this mean?). If published, this will include your full peer review and any attached files.

Reviewer #1: **Yes: **Doug Speed

Reviewer #2: No

Reviewer #3: No

---

## [Decision Letter · Decision Letter 1]

7 Aug 2023

Dear Dr Briollais,

We are pleased to inform you that your manuscript entitled "Large-scale whole exome sequencing studies identify two genes,CTSL and APOE, associated with lung cancer" has been editorially accepted for publication in PLOS Genetics. Congratulations!

Yours sincerely,

Vincent Plagnol

Academic Editor

PLOS Genetics

David Kwiatkowski

Section Editor

PLOS Genetics

Comments from the reviewers (if applicable):

Reviewer's Responses to Questions

**Comments to the Authors:**

Reviewer #1: The authors have addressed all my comments, which included adding a permutation analysis. Thank you

Reviewer #2: My comments have been addressed.

Reviewer #3: I really appreciate the authors' efforts to polish the manuscript and address all reviewers' comments. The overall quality is much higher than the old version. I only had one minor comment: It will be great if the authors can compare their results with more popular methods, such as SAIGE-GENE/ SAIGE-GENE+, which can be served as additional validation. If they had similar findings, then the results will be more convincing. Thanks.

**Have all data underlying the figures and results presented in the manuscript been provided?**

Reviewer #1: Yes

Reviewer #2: None

Reviewer #3: Yes

PLOS authors have the option to publish the peer review history of their article (what does this mean?). If published, this will include your full peer review and any attached files.

Reviewer #1: **Yes: **Doug Speed

Reviewer #2: No

Reviewer #3: No

**Data Deposition**

http://datadryad.org/submit?journalID=pgenetics&manu=PGENETICS-D-22-01475R1

**Press Queries**

---

## [Editor Report · Acceptance letter]

29 Aug 2023

PGENETICS-D-22-01475R1 

Large-scale whole exome sequencing studies identify two genes,CTSL and APOE, associated with lung cancer 

Dear Dr Briollais, 

We are pleased to inform you that your manuscript entitled "Large-scale whole exome sequencing studies identify two genes,CTSL and APOE, associated with lung cancer" has been formally accepted for publication in PLOS Genetics! Your manuscript is now with our production department and you will be notified of the publication date in due course.

With kind regards,

Livia Kovacs

PLOS Genetics

On behalf of:
